# CO_2_ Breathing Prior to Simulated Diving Increases Decompression Sickness Risk in a Mouse Model: The Microbiota Trail Is Not Forgotten

**DOI:** 10.3390/ijerph21091141

**Published:** 2024-08-28

**Authors:** Lucille Daubresse, Aurélie Portas, Alexandrine Bertaud, Marion Marlinge, Sandrine Gaillard, Jean-Jacques Risso, Céline Ramdani, Jean-Claude Rostain, Nabil Adjiriou, Anne-Virginie Desruelle, Jean-Eric Blatteau, Régis Guieu, Nicolas Vallée

**Affiliations:** 1Service de Médecine Hyperbare, Hôpital d’Instruction des Armées, 83000 Toulon, Francejean-eric.blatteau@intradef.gouv.fr (J.-E.B.); 2Université de Toulon, 83130 La Garde, Francesandrine.gaillard@univ-tln.fr (S.G.); 3Aix-Marseille University, 27 Boulevard Jean-Moulin, 13005 Marseille, Francejean-claude.rostain@univ-amu.fr (J.-C.R.); nabil.adjiriou@univ-amu.fr (N.A.); regis.guieu@univ-amu.fr (R.G.); 4Subaquatic Operational Research Team (ERRSO), Military Institute of Biomedical Research (IRBA), 83000 Toulon, Franceceline.ramdani@intradef.gouv.fr (C.R.); anne.desruelle@intradef.gouv.fr (A.-V.D.)

**Keywords:** dysbiosis, Akkermansia, Rhizobium, gut–brain axis, hypercapnia

## Abstract

Decompression sickness (DCS) with neurological disorders is the leading cause of major diving accidents treated in hyperbaric chambers. Exposure to high levels of CO_2_ during diving is a safety concern for occupational groups at risk of DCS. However, the effects of prior exposure to CO_2_ have never been evaluated. The purpose of this study was to evaluate the effect of CO_2_ breathing prior to a provocative dive on the occurrence of DCS in mice. Fifty mice were exposed to a maximum CO_2_ concentration of 70 hPa, i.e., 7% at atmospheric pressure, for one hour at atmospheric pressure. Another 50 mice breathing air under similar conditions served as controls. In the AIR group (control), 22 out of 50 mice showed post-dive symptoms compared to 44 out of 50 in the CO_2_ group (*p* < 0.001). We found that CO_2_ breathing is associated with a decrease in body temperature in mice and that CO_2_ exposure dramatically increases the incidence of DCS (*p* < 0.001). More unexpectedly, it appears that the lower temperature of the animals even before exposure to the accident-prone protocol leads to an unfavorable prognosis (*p* = 0.046). This study also suggests that the composition of the microbiota may influence thermogenesis and thus accidentology. Depending on prior exposure, some of the bacterial genera identified in this work could be perceived as beneficial or pathogenic.

## 1. Introduction

Exposure to high levels of CO_2_ is a growing safety concern for professional divers at risk of decompression sickness (DCS) [1,2,3]. In fact, pre-dive exposure to CO_2_ is suspected in a recent series of unusual DCS cases with neurological symptoms [4]. Although there is a tendency for hypercapnia during diving [2] and an increase in osteo-articular accidents among hyperbaric workers breathing a CO_2_-enriched gas mixture (1.8 to 2.3%) [5], there is a paucity of information on the effects of CO_2_ inhalation prior to hyperbaric exposure and its incidence in DCS [6].

At atmospheric pressure, the increase in blood CO_2_ levels, endogenous or exogenous, defines hypercapnia, which in humans corresponds to an arterial CO_2_ partial pressure (PaCO_2_) greater than 45 mmHg (60 hPa). This causes a very small decrease in plasma pH, which limits the association of oxygen with hemoglobin at the pulmonary level and promotes its dissociation at the systemic level. In response, the body increases its respiratory rate and essentially mobilizes its bicarbonate buffering systems according to Le Chatelier’s principle [7,8] (CO_2_ + H_2_O⇔H_2_CO_3_⇔HCO_3_^−^ + H^+^) to eliminate excess CO_2_ and keep the blood pH stable.

According to the classical scheme, DCS results from the formation of bubbles in the body [9]. The tissue gas load in the body that occurs during compressed air ventilation during diving is proportional to the duration and depth of the dive. Bubble formation occurs during desaturation, i.e., ascent to the surface to sea level, when gas elimination by perfusion and ventilation is too inefficient. Bubbles that “remain trapped” in tissues exert their effects primarily through compression of their surroundings. In the bloodstream, the presence of bubbles (i.e., vascular gas embolism) causes cell destruction by increasing shear forces [10] but also platelet activation, possibly due to direct bubble/platelet contact [11,12,13], ultimately causing inflammation, vasoconstriction, circulatory stasis, and interstitial edema, leading to tissue death [14]. Disorders of the spinal cord and brain are the most serious symptoms treated at the hyperbaric center. They can lead to paralysis, with a risk of long-term neurological sequelae of about 20 to 30 percent [15].

Circulating bubbles are essentially composed of inspired gases such as nitrogen [16], but also metabolized gases. A porcine model of a desaturation accident specifies that the composition of the bubbles includes 5 to 9% CO_2_ and 1 to 4% O_2_, while the remainder of the bubble is inert gas [16]. A Japanese study conducted on rabbits highlights that the CO_2_ level in the bubble seems to decrease first and then gradually increase, while the O_2_ level evolves in the opposite way [17]. The concentration of CO_2_ in the bubble and its evolution with time, including counter-diffusion [18], could play an important role in the growth and formation of bubbles during decompression [19,20]. We are not currently aware of any studies that have examined the effect of adding CO_2_ on the incidence of DCS. The suggestion [21] that CO_2_ retention increases the likelihood of DCS is based on the assumption that tissue gas uptake is enhanced by an elevated pCO_2_ (CO_2_ partial pressure) during deep work and is proportionally slower at a normal pCO_2_ during decompression. In fact, such CO_2_ toxicity is often associated with oxygen toxicity of the nervous system [22,23,24] and possibly blood pH [20]. Because CO_2_ is a highly diffusible gas, we hypothesize that its increase in blood concentration could significantly contribute to increased bubble growth and therefore DCS.

Inflammation, or even autoimmune reaction [25,26], is often transcribed in articles relating to DCS [27,28,29,30]. Differences have thus been observed in animals all from the same strain [28] or in specially selected strains [25,31,32]. This naturally led the teams to focus on the predispositions that the host microbiota can induce [25,30,33,34,35]. The microbiota is indeed known for its interactions with the host, for example by modulating immune responses [25,27,36,37,38,39] or/and thermoregulation [40,41,42,43,44]. The microbiota is sensitive to temperature fluctuation [41] and can confer host plasticity in thermoregulation [37,40]. As an example, disruption of the equilibrium of the gut microbiota by cold and a high-fat diet can compromise the integrity of the intestinal barrier such that it induces increased permeability, facilitating the translocation of harmful substances and bacteria into the bloodstream and ultimately triggering systemic inflammation [45].

On the other hand, increases in CO_2_ are also known to modulate the microbiota [46], as well as the inflammatory response [46,47] and thermoregulation [19,46,48]. Thermogenesis is the cornerstone of metabolism [49]. Thermoregulation is closely related to the metabolic function and health of the host [36]. Hypercapnia elicits hypothermia in a great number of vertebrates, including mammals [50]. This mechanism could include peripheral and central vasomotion through NO activation, and also hyperventilation, which would lead to an increase in pulmonary perspiration, among other things, and therefore to the cooling of the body, especially during exercise [19,51,52].

The purpose of this study was to evaluate the effect of CO_2_ breathing prior to a provocative dive on the occurrence of DCS. This work is the first to clearly establish the influence of exposure to a high concentration of CO_2_ on the risk of DCS. This study was performed in a mouse model with a clinical and biological evaluation, including thermal measurements and characterization of the native fecal microbiota. The hyperbaric protocol was preceded by exposure of the mice to a maximum CO_2_ concentration of 70 hPa, i.e., 7% at atmospheric pressure, for one hour. This CO_2_ value corresponds to the maximum concentration that does not cause clinical or social effects in the animals [53,54,55].

## 2. Materials and Methods

The experimental design can be followed in Figure 1. This study was conducted in Marseille (France). On the day of the experiment, the surface temperature of each mouse was measured. The mice were then immediately identified (with a colored barcode affixed to the base of the tail), weighed, and their feces collected for microbiota analysis. Next, the mice were exposed to either CO_2_ or air. After one hour of exposure, the cages were transferred to the hyperbaric chamber. After the accident-prone protocol, a clinical examination supplemented by temperature measurement was performed. Blood punctures were then performed under anesthesia for biochemical analysis. See details in the following paragraphs.

### 2.1. Ethics Statements

All procedures involving animals complied with European (Directive 2010/63/EU) and French (Decree 2013/118) legislation. The protocol was approved by the Animal Ethics Committee (CE14, University of Aix-Marseille) on 6 May 2022 under number APAFIS#34720-2022011812031043. The population consisted of 100 6-week-old male C57Bl/6n mice (Charles River strain, The Jackson Laboratory, Italy). Animals were housed 5 per cage (as delivered) in an accredited animal facility with a controlled temperature (22 ± 1 °C) and day/night cycle (12 h of light per day, 7:00 am–7:00 pm), with ad libitum access to water and food (A03, UAR).

### 2.2. Animal Batches

The animals remained grouped in batches of 5, as they were when delivered from the breeder, to maintain social grouping. Each batch of 5 mice was assigned to one of two groups. The first group consisted of 50 mice exposed to a CO_2_-enriched atmosphere (CO_2_), while the control group (AIR) consisted of 50 mice breathing air in a similar condition. After one hour of exposure, the animal cages were transferred to the hyperbaric chamber.

### 2.3. Evaluation of Superficial Temperature by Infrared Imaging

The surface temperatures of the mice were measured twice by infrared thermography (camera: model Flir E52, Sweden), at the very beginning of the experiment and before anesthesia. Infrared videos were taken in a room at 23 °C while the mice were moving quietly in their cages. More specifically, the animals were filmed in their original cage before being weighed and tagged, then transferred to a temporary cage, and then all returned to the original cage. The same strategy was used before anesthesia. The images were then analyzed to evaluate the averaged temperature using the processing software of the camera manufacturer (Flir ResearchIR, Wilsonville, OR, USA), according to our previous study [28].

### 2.4. CO_2_ Exposition

A constant flow of CO_2_, controlled by an Arduino system designed by our engineer, perfused the polycarbonate cage to maintain a 7% CO_2_ atmosphere for 1 h. The same cage perfused with air served as a control. The animals were then immediately transferred to the hyperbaric chamber.

### 2.5. Hyperbaric Exposure

Batches of 10 mice (5 per cage and 5 per group) from the AIR and CO_2_ pools were subjected to the hyperbaric protocol in an 80 L tank equipped with two observation ports. Mice were allowed to move freely within their cages.

The air compression protocol involved two ramps of pressure increase, first at 0.1 atm/min to a pressure of 2 atm, followed by 1 atm/min to 10 atm; 10 atm (equivalent to a depth of 90 msw) is the pressure at which the animals were maintained for 45 min prior to decompression. The decompression rate was 0.1 atm/s to 1 atm.

In the hyperbaric chamber, CO2 and water vapor produced by the mice were trapped with soda lime (<300 ppm) and silica gel (relative humidity: 40–60%). The gases were mixed by an electric fan. The day–night cycle was maintained throughout.

### 2.6. Physical Examination

After exiting the hyperbaric chamber, a 30 min period was devoted to the identification of clinical signs of a desaturation accident, with the addition of grip tests at 15 and 30 min [28,30,56,57]. Manifestations of DCS include itching, respiratory dysfunction (gasping), neurological symptoms (tremors, convulsions, motor and/or sensory deficits including claudication and paralysis), marked asthenia (decrease of gripping duration, prostrate mouse without exploratory posture), and in the most severe form, it may result in the death of the animal [28,30,56,57].

### 2.7. Anesthesia and Biopsy

After physical examination, the animals were first anesthetized by administration of gaseous isoflurane (4.0% in oxygen flow at 2.0 L/min) in an individual cage for 2 min and then through a face mask (anesthesia box: Harvard Apparatus, Les Ulis, France). The intraperitoneal injection of ketamine (100 mg·kg^−1^) and xylazine (10 mg·kg^−1^) completed the anesthesia. The degree of anesthesia was determined by testing the absence of withdrawal reflexes in response to pinching of the distal hind limbs.

Immediately after anesthesia, blood samples were collected by intracardiac puncture for assays of glial-derived peptide S100 (PS100), copeptin, neuron-specific enolase (NSE), bicarbonate, and interleukin-6 (IL-6). Blood was collected in a sterile 1 mL syringe containing lithium heparin, immediately placed on ice, and then processed according to the manufacturer’s instructions. Finally, the animals were euthanized with a lethal dose of anesthetic.

### 2.8. Blood Biochemical Analysis

Plasma was diluted volume-to-volume to obtain a sufficient sample volume for all biochemical analyses. All analyses were performed in duplicate. The coefficient of variation tolerated by the analyzers had to be less than 10% for the measurement to be considered valid.

Glial-derived peptide S100 (PS100) analysis was performed by electrochemiluminescence on the Cobas e411 analyzer (Roche^®^, Meylan, France; detection limit 5 pg/mL). A total of 20 µL of plasma was incubated with 2 different antibodies specifically directed against PS100: a monoclonal and biotinylated antibody and a second antibody that binds to ruthenium to form a sandwich complex. Magnetic microparticles coated with streptavidin were added to bind the immune complex through biotin and to keep it in the measuring cell despite a magnetic field washing step. A potential difference was then applied to induce luminescence due to the excitation of ruthenium (quantified by a photomultiplier), this signal being proportional to the concentration of PS100 present in the sample.

Copeptin (50 µL per assay/detection limit 0.88 pmol/L) and NSE (70 µL per assay/detection limit 4 ng/mL) were analyzed on a Kryptor Gold instrument (Thermoscientific^®^, Hillsboro, OR, USA) using TRACE^®^ (Time-Resolved Amplified Cryptate Emission) technology. Briefly, the analyzer measures the fluorescence at 620 and 665 nm emitted by an immune complex-containing donor/acceptor system (and europium cryptate and cyanine, respectively) after an excitation step at 337 nm.

Bicarbonate (HCO_3_^−^) was determined with a volume of 5.7 µL per assay on an Atellica Solution (Siemens^®,^ Munich, Germany) platform equipped with the CH930 module (detection limit of 10 µmol/L). The assay uses a colorimetric enzymatic method based on two chemical reactions, catalyzed by phosphoenolpyruvate carboxylase (PEPC) and malate dehydrogenase (MDH). The reduction in absorbance is proportional to the concentration of the analyte. Similarly, IL-6 (50 µL per assay/detection limit 2.7 pg/mL) is determined on the same platform with the IM module, which employs chemiluminescence using a mouse monoclonal antibody labeled with an acridinium ester molecule and monoclonal antibody-coated magnetic microparticles, both directed against IL-6.

### 2.9. Stools Bacterial Content: 16s rDNA Extraction and Sequencing

While the mouse clung to the grid during marking, feces falling onto a new surgical field were collected with disinfected tweezers, placed in a 1.5 mL DNA-free Eppendorf tube, and stored at −80 °C until genetic analysis. Deoxyribonucleic acid (DNA) for the metagenomic identification of microbiota was extracted from stool using the Maxwell RSC Fecal Microbiome DNA Kit (Maxwell RSC, Promega, Charbonnières-les-Bains, France) according to the supplier’s recommendations to obtain a 100 µL eluate. The concentration of the extract was adjusted to 20 ng/µL in ultrapure water prior to amplification on the Azenta Genewiz platform (France). The 16S-EZ workflow involves a proprietary multiplexed Polymerase Chain Reaction (PCR) amplification of the V3 and V4 hypervariable regions of the 16S rDNA gene. A limited-cycle 2nd round PCR adds sample-specific barcodes to each sample to allow the multiplexing of multiple samples on the same sequencing run. Final libraries are pooled and undergo final quality control prior to sequencing. Sequencing is performed using Illumina chemistry in the 2 × 250 bp paired-end configuration to ensure coverage of all V3 and V4 sequences. Subsequent sequencing data undergo quality control prior to 16S-EZ bioinformatics analysis.

### 2.10. Bioinformatic Processing

The initial quality control of reads in ‘fastq’ files was performed using FastQC [58]. Forward and reverse primers were trimmed from the paired-end reads. Sequences were quality filtered, denoised, and merged into amplicon sequence variants (ASVs) using DADA2 [59] within R [60] (v. 4.1.0). Briefly, sequences were filtered based on a visual assessment of error rates according to the quality score ≥ 25 and after chimeras were detected and removed using the ‘removeBimeraDenovo’ script in DADA2. Each representative 16s rDNA ASV sequence was associated with Blast+ (v. 2.10) using FROGS v.4.0.1 [61] and the naive Bayesian classifier method [62] against the SILVA 138 database [63]. Based on the initial assignment, we organized our taxonomic profile to retain the taxonomy of ASVs with greater than 95% identity and 75% sequence coverage. Below these thresholds, if an ASV had a bootstrap confidence score < 65% at any level, it was binned as unknown. More specifically, if the bootstrap confidence score associated with the ASV was less than 65% at any level of classification (e.g., genus, family, etc.), then that ASV was labeled as belonging to an unknown species per example. After processing, ASVs with a frequency lower than 0.005% were removed [64]. The raw sequence data generated in this study have been deposited in the NCBI Sequence Read Archive under the accession number BioProject PRJNA1070862.

### 2.11. Statistical Analyses

Prior to analysis, a normality test was conducted. Data normalization (n) or non-parametric tests were applied when necessary.

The association between qualitative (gas exposure, clinical status) and quantitative variables (weight, temperature, bacterial species abundances…) was studied using Partial Least Squares Regression–Discriminant Analysis (PLS-DA) [65]. It is based on the Partial Least Squares method and allows one to handle multicollinear data, missing values, and data sets with few observations and many variables. The PLS-DA was set with press minimum as the stop criterion, and the jackknife resampling technique was selected for cross-validation. The variable importance for projection and discriminative effects of gas or clinical status were evaluated.

Kruskal–Wallis (KW) tests supplemented with Dunn’s post hoc (with Bonferroni correction) were performed for qualitative or non-parametric data [66].

The software used was XlStat Biomed from Addinsoft (Bordeaux, France). The maximum alpha level accepted was 5%.

## 3. Results

### 3.1. Clinical Status of the Animals Following the Dive

The initial results confirmed that the protocol was accident-prone as expected.

Clinical observation after the simulated dive was the main criterion for defining DCS. During the thirty-minute observation period following exit from the hyperbaric chamber, 66% (out of 100 mice) of the total animals showed symptoms, including convulsions and paralysis in the most severe cases. Two mice died, one per group. There was a significant difference (KW; *p* < 0.001) between the group of mice that breathed air before the simulated dive and the group that breathed 21% O_2_ and 7% CO_2_ (Figure 2). In the AIR group (control), 22 out of 50 mice showed post-dive symptoms compared to 44 out of 50 in the CO_2_ group.

Knowing that weight can influence accidents, we tested it: no significant difference between the groups was found (mean weight ± SD, 19.2 ± 1.7 g; KW, *p* = 0.239).

### 3.2. General Description of the Data

A PLS-DA was conducted to model the interaction between the 42 variables studied (see Figure 4 for the complete list), with clinical assessment as an additional qualitative variable. Previous exposure to CO_2_ or its control in air was integrated as the only explanatory qualitative variable. The model showed a good fit, with R^2^X approaching 1 as the number of components increased. R^2^Y, while slightly slower, also demonstrated a good fit. However, the model had limited predictive ability, indicated by a low Q^2^ value. However, the Receiver Operator Characteristic (ROC) curve analysis showed an area under the curve (AUC) of 0.879, with specificity and sensitivity values of 79% and 80%, respectively, resulting in an accuracy of 80%.

Projection (Figure 3 left) onto components 1 and 2 explains 35.9% of the model, with component 1 alone expected to explain 31%. The confidence ellipses (Fisher test, 95% CI) surrounding the observations of symptomatic and asymptomatic animals on the PLS-DA indicate a slight discrimination of these groups on the T1 axis. The confidence ellipses surrounding the animals according to inhaled gas have a profile very similar to that of their clinical status.

The correlations (Figure 3 right) between the components and the variables are generally good, with a considerable number of variables projected onto the disk between 0.8 and 1, including, for example, the nature of the gas, which is essentially projected onto axis t1, while a number of bacteria are projected onto the t2 axis. In general, the correlations are weak in the center. For example, we can see the weight of the animals close to the center. In this particular case, we had previously selected animals of low weight to overcome this well-known problem in diving, where a high weight increases accidents.

Looking specifically at the correlations and the projection of the clinical axis (Sympto/Asympto, green line), we note with interest a positive correlation between an asymptomatic state and prior exposure to air (as a control), as well as the higher temperature of the mice upon exiting the hyperbaric chamber. This supports the diagnosis. To a lesser extent, and because it is clearly on the Sympto/Asympto axis, the PS100 marker could be considered. It may also appear that the higher temperature of the mice before the start of the experiment is positively correlated with being asymptomatic. The opposite would be true in the case of prior exposure to CO_2_, which would be clinically deleterious and would be associated with an increase in bicarbonate and NSE.

Study of the classification of the variables of importance for the projection (VIP) of the model axes confirms the relationship between the previous exposure to the gas (VIP1 = 3.354) and the clinical status, accompanied by the temperature on leaving the hyperbaric chamber (VIP1 = 2.757), and also the temperature before the start of the experiment (VIP1 = 1.554). PS100 (VIP1 = 1.260), NSE (VIP1 = 0.853) and bicarbonates (VIP1 = 0.744) are also important data. These values decrease slightly on projection axes 2 and 3. This is followed by *Muribaculaceae* (unidentified genus) (VIP1 = 1.090; VIP2 = 1.017), *Lachnospiracée_GCA-9000066575* (VIP1 = 0.892; VIP2 = 0.881), and *Christensenellaceae* R-7 group (VIP1 = 0.846; VIP2 = 0.941), in decreasing order of importance on projection axis 1. Globally, microbes are more abundant on axes 2 and 3. However, the importance of these bacterial genera, at t1 and t2, underlies the link between the initial presence of these bacteria and the clinical status, taking into account the value of the contribution of axis 2 (4.9%) compared to axis 1 (31%) for the construction of the model.

The discriminant analysis based on the values of the variables of importance in the projection according to the discrimination (VIDs) specifies the interest of the parameters according to the clinical state (Figure 4). Thus, the Asympto state is more clearly associated with a positive correlation with previous exposure to air (rather than CO_2_), a high surface temperature, PS100 and IL-6 values, and also with the presence of *Oscillospiraceae_intestinimoneas*, *Nostocaceae_Tolypothrix*, *Akkermansiaceae_Akkermansia,* et cetera. Negative correlations compared to the Asympto status can also be seen in the figure, which ultimately bring us closer to the Sympto status. On the other hand, the symptomatic status is positively correlated with the previous ventilation of CO_2_ and increased bicarbonate and NSE levels (which would support the clinical status), but also with the initial presence of unidentified genera of *Muribaculaceae*, *Roseburia* and *Lachnospiraceae* in the stool. The ratio of aerobic/anaerobic strains seems less interesting for model projection, which does not mean that it is not clinically significant. Post hoc tests then specified the significance of each criterion.

#### 3.2.1. Two-Way Analysis

There are a number of variables that catch our attention when reading the PLS-DA.

##### Body Temperature

Infrared thermograms performed after the hyperbaric protocol (KW; *p* < 0.001) show differences between the clinical groups. In this case, symptomatic animals previously exposed to CO_2_ had a lower mean surface temperature (~30.9 °C vs. ~32.0 °C) than mice in other groups (Figure 5).

Even before the start of the protocol, the temperature of the animals (Figure 5) would tend to be determinative, to the extent that the symptomatic animals had different temperatures than the uninjured animals (KW; *p* = 0.046). It should be noted that the initial temperature of symptomatic AIR mice tended to be lower (~0.4 °C) than that of asymptomatic AIR mice (post hoc Dunn test; *p* = 0.0087). This point is consistent with the PLS-DA, i.e., with high VIP and VIDs scores.

##### Plasma Analysis

Plasma variables such as IL-6, NSE, copeptin, bicarbonate, and PS100 measured after the accident-prone protocol do not show significant differences between the groups (Figure 6). There is a trend for NSE, especially between the Asympto_CO_2_ mice with lower levels than the Asympto_AIR mice.

#### 3.2.2. Initial Fecal Content Analysis

A 16s rDNA analysis was performed on mouse feces collected at the very beginning of the protocol. Approximately 97% (min 4%–max 99%) of ASVs were not classified/identified, so these results must be interpreted with caution (see Limitations section). Finally, 520 ASVs from the stools of 88 mice allowed the identification of about thirty genera, distributed in about 13 families and 6 classes, with the presence of unknowns (Figure 7). Only about fifteen species appeared to be correctly identified, prompting analyses at the genus level. The distribution of relative abundances (Figure 7, right side) according to Bray–Curtis dissimilarity agglomerative hierarchical clustering (ACH; Figure 7, left side) on the genera allows us to separate groups without them directly corresponding to our clinical groups. The same is true for richness indices based on ASVs (Figure 7, middle). A plot of diversity according to the Shannon index is available in the supplementary data (Appendix A). These results should be viewed in the context of the contribution of bacteria to the PLS-DA model.

Cluster I of the ACH, with a high proportion of symptomatic animals (8/10), presents a relatively high richness (Shannon diversity index close to 3.5), but with a lot of Clostridia. This cluster also has a very low proportion of aerobic genera (min 0%–max 2%), while the average for all animals is 35 ± 35%. On the other hand, cluster II, consisting of eight symptomatic mice (/11), has a very high aerobic presence (min 97%–max 100%). This cluster, whose Shannon index is close to 3.0, shows a particular dominance of two families, *Sphyngomonadacea (Alphaproteobacteria)* and *Nostoccaceae (Cyanobacteria)*. Cluster IV, with five symptomatic mice (/10), has the lowest microbial diversity (Shannon close to 1) and a median aerobic rate of 16% (min 0%–max 66%). Cluster VIII, with eight symptomatic animals (/10), has a dominance of *Rhizobiaceae (Alphaproteobacteria)* and *Cyanobacteria Nostocaceae* (Shannon close to 2.1) and a median aerobic rate of 84% (min 16%–max 98%).

Overall, Shannon, coverage or Pielou evenness, and dominance dmn (McNaughton’s dominance index) indices (Appendix A) organized by clinical and gas groups remain non-significant (KW; *p* > 0.05), although at the post hoc level, trends (Dunn; *p* < 0.05) appear between air-exposed mice (symptomatic versus asymptomatic). Asymptomatic animals appear to have lower Shannon diversity indices.

A more specific analysis of genera reveals differences between groups, in relative or raw abundances (Figure 8 and Appendix A).

Significant differences, both in relative and raw abundances, are highlighted in the genera *Blautia* and *GCA-900066575* of the family *Lachnospiraceae*, with differences at the post hoc level. There are much higher values of *Lachnospiraceae Blautia* in the Sympto_AIR compared to the other groups, especially compared to Sympto_CO_2_, where *Blautia* is almost absent.

There are significantly higher values of Lachnospiraceae GCA-900066575 in Sympto_AIR compared to Asympto_AIR. A similar trend is found at the post hoc level for *Candidatus Saccharimonas*, with higher values in Sympto_AIR compared to Asympto_AIR.

From a less strict statistical point of view, we see trends (KW with *p* < 0.010) in Allorhizobium–Neorhizobium–Pararhizobium–Rhizobium, whose values are higher in Asympto_CO_2_ compared to Asympto_AIR and Sympto_AIR. The genus Nostoc PCC-73102 has slightly higher values in Asympto_AIR compared to Sympto_AIR (without relative abundance playing a role). Roseburia has higher abundance values in Sympto_AIR compared to Asympto_AIR and Sympto_CO_2_. The genus Anaerostripes has higher relative (and raw) abundance values in Asympto_CO_2_ compared to Asympto_AIR.

## 4. Discussion

The purpose of this study is to evaluate the effect of CO_2_ breathing prior to a provocative dive on the occurrence of DCS. To our knowledge, this work is the first to demonstrate that prior exposure to a CO_2_-enriched atmosphere increases the risk of DCS in mice. Although thermograms taken after the hyperbaric protocol seem to support the diagnosis, more generally the temperature of the animals is associated with the clinical status, without us knowing if it is the consequence or even the cause. According to the PLS-DA, the variation in bicarbonate levels in the blood confirms low adaptive CO_2_ exfiltration after the provocative protocol, especially after a previous CO_2_ exposure. This would require the use of a buffering system [67].

The trend of differences in NSE levels between CO_2_- and AIR-exposed mice may suggest the involvement of different pathogenic pathways, where neurons seem to be less affected in Asympto_CO_2_ mice.

A specific bacterial community, based on diversity, does not appear to contribute collectively to preventing or increasing DCS risk. However, some bacteria, for example from the *Lachnospiraceae* family, appear to play a role depending on whether or not there is prior exposure to CO_2_. Although secondary, this suggests that more than one pathway may alter the outcome.

### 4.1. Effects of CO_2_ and Diving on Circulating Markers of Neuronal Stress

Neuron-specific enolase (NSE) and the glial-derived peptide S100 calcium-binding protein B (S100B) are commonly described to assess the presence and severity of neurological injury. S100B is known to rise in the peripheral circulation in association with both oxidative [68] and heat stress [69], and a relationship is described with core body temperature (Tc) response [70]. NSE is a glycolytic enzyme found in brain neurons and peripheral nervous tissue that is elevated during cerebral inflammation or ischemia. These proteins were measured in neurological diving accident victims (symptomatic) and compared with asymptomatic divers (same diving profile). In fact, the level of NSE is higher in the symptomatic group than in the control group [71], and the S100B level shows no change [71]. In rat studies, there appears to be no change in plasma S100B concentrations after a low-bubble-grade dive, and it is increased after a high-bubble-grade dive [72,73]. Interestingly, S100B protein levels increase with apnea-induced hypoxic stress [74].

In our study, S100B remains non-contributory. S100B does not appear to be sensitive to cold (see below). Differences in NSE levels between CO_2_- and AIR-exposed mice may indicate that neurons in Asympto_CO_2_ mice appear to be less affected, but we cannot explain why this difference is not found between the symptomatic and asymptomatic AIR groups. Caution should therefore be exercised in interpreting these data. However, it is one avenue that would be worth exploring, given the high rates of NSE expressed in patients with chronic hypercapnia and CO_2_ retention [75].

### 4.2. Effects of CO_2_ and Diving on Plasma Markers, Thermoregulation, and Vasomotion

A high level of bicarbonates over time, as supported by the PLS-DA, could affect cerebral vasomotor function and blood flow. Hypothetically, this vasodilation could make the brains of mice previously exposed to CO_2_ more susceptible to bubbles (see the Moderation section), both by increasing gas saturation and by making it more accessible to bubbles of systemic origin.

### 4.3. Temperature

In the present study, the thermograms performed after the hyperbaric protocol seem to confirm the diagnosis, as expected according to previous studies [28]. Symptomatic mice have a lower surface temperature (~1.1 °C). To explain this low temperature in symptomatic mice, it would be simplistic to exclude a pro-inflammatory state (usually associated with temperature elevation) as a hypothesis, especially in CO_2_-exposed mice. This should probably affect vasomotion, the quality of tissue perfusion, i.e., gas balance and bubble growth, according to the most common theory on DCS [9].

More unexpectedly, it should be noted that the initial temperature (before any exposure) of symptomatic AIR mice tends to be lower (~0.4 °C) than that of asymptomatic AIR mice. This suggests that initial temperature is critical for prognosis, and that a lower baseline temperature currently predicts a state of susceptibility to DCS. Considering that hypercapnia induces hypothermia, but sufficient heat production is a prerequisite for inflammation, a special study should be devoted to it. Again, we might wonder about the links between NO/vasodilation/thermogenesis [76] and now CO_2_ in DCS.

### 4.4. Impact of the Microbiota in the Stress Response

The gut microbiota has been of interest to hyperbaric teams for several years. It has been documented that the core of the intestinal contents is hypoxic, but there is a gradient of oxygen concentration near the intestinal epithelium [77], and oxygen or hypoxia can reshape its community [78,79]. For example, an increase in the frequency of the Clostridium group is shown after a safety dive in men [35].

In this study, the collection of feces at the beginning of the protocol and the analysis of 16s DNA were conducted to investigate whether the microbial profile could play a critical role in the response to the accident-prone protocol. While no specific bacterial spectrum was identified, it was found that certain bacteria, when examined individually, appeared to have an impact on clinical outcomes depending on whether or not there was prior exposure to CO_2_. This suggests that while CO_2_ may play an important role in symptoms, the gut microbiota may also have an indirect influence, although this relationship is less pronounced according to the PLS-DA analysis.

The use of richness indices, such as the Shannon index, did not allow for the definition of a specific risk fingerprint. This may may be attributed to the use of an inbred animal strain, C57Bl6/J, which exerts a selective pressure on the gut microbiota, resulting in low diversity as reported by the Shannon index. This limitation or advantage of the study is noteworthy because reducing diversity enhances the contrast, making it easier to identify the microorganisms involved in pathology. This finding supports the notion of inter- and intra-individual variability that has been described in both animal and human diving studies [25,33,34]. However, it should be noted that some bacterial genera demonstrated intergroup differences in abundance, as indirectly suggested in previous studies [80,81,82]. For example, the low representation of *Lachnospiraceae_GCA-900066575* in this study appears to have a beneficial effect after a risky dive, as the Asympto_AIR animals showed significantly less abundance of this bacterium compared to their symptomatic counterparts (i.e., those not exposed to CO_2_).

Certain bacteria require further caution or additional targeted studies. For instance, *Candidatus_Saccharimonas* is significantly more abundant in Sympto_AIR animals compared to Asympto_AIR animals. However, its role as a bacterium promoting DCS may be debated, since its contribution to the model was very weak. Similarly, Nostoc_PCC-73102 demonstrated a slightly greater contribution to the model but only displayed a trend towards being pathogenic. The higher abundance of *Lachnospiraceae_Roseburia* in Sympto_AIR mice suggests a potential promotion of DCS, while its absence may be beneficial in the event of a provocative dive. Paradoxically, this same low abundance may favor DCS in the presence of prior CO_2_ inhalation. Similar trends were observed for *Lachnospiraceae_Blautia*, with higher levels of significance in their abundance (*p* < 0.002), but low contribution values to the model. Furthermore, a higher initial proportion of *Lachnospiraceae_Anaerostripes* tends to be beneficial in the case of prior exposure to CO_2_, despite its relatively low contribution to the model. These results contribute to the ongoing debate regarding the role of *Lachnospiraceae* in physiology, as they are known to be major producers of short-chain fatty acids [83]. Further investigations are needed to clarify the specific actions of the Lachnospiraceae family observed in this study. We carried out tests at the level of the Lachnospiraceae family, which did not reveal any significant differences. This implies a more specific action that remains to be clarified. In fact, the effect of short-chain fatty acids (SCFA) by fermentation has already been mentioned in DCS, as a protector but also as a promoter of accidents [33,81,84,85,86].

Recent work suggests that the microbiota plays an increasingly important role in thermoregulation through its metabolic derivatives [42] or through its serotonergic effects via modulation of the gut–brain axis [36,37]. For instance, the inhibition of serotonin reuptake with fluoxetine has been shown to reduce DCS [29,30], and it would be interesting to explore this question further through the influence of the microbiota. The influence of temperature in DCS is discussed earlier in this document, and we could now add a possible role for bacteria. It should be remembered that the fecal samples were taken at the very beginning of the protocol. Recent studies have shown that cold exposure can lead to dramatic changes in the composition and function of the gut microbiota, with acute responses, and *Akkermansia muciniphila* may be the most appropriate strain [43,44]. Looking at the VIDs ranking, the amount of Akkermansia (*Akkermansia muciniphila* is the only one identified in this genus) could be positively correlated with lower accident rates or, on the contrary, negatively correlated with prior exposure to CO_2_.

Among the noteworthy findings, a higher initial proportion of Rhizobiaceae–Allorhizobium–Neorhizobium–Parhizobium–Rhizobium (*Rhizobium* sp. is the only one identified in this genus) would tend to be beneficial in cases of prior CO_2_ exposure, despite its relatively low contribution to the model. Allorhizobium–Neorhizobium–Pararhizobium–Rhizobium is related to nitrogen fixation by plants. Therefore, it could come from the diet of mice. These bacteria have been proposed to be involved in the development of multiple sclerosis, where their consumption from root bulbs containing Rhizobiaceae acts as exogenous antigens that trigger an immune response [38]. These bacteria are also known to neutralize the immune response [39], and their role in DCS warrants further exploration based on the findings of this study and others [25,27].

### 4.5. Context

This study was particularly motivated by an unusual accident where high levels of CO_2_ were detected. When fresh air is around 0.04% CO_2_, wearing face masks for more than 5 min carries a possible chronic exposure to carbon dioxide of 1.41% to 3.2% of the inhaled air [87,88,89]. In this context, relatively speaking, this work could open the question of wearing a face mask. In fact, in a situation such as the COVID epidemic, where the rules regarding the wearing of masks can be particularly strict, special attention should be paid to professional divers.

### 4.6. Moderation, Study Limits

The mouse is probably not the best model for DCS, and as with any model, its limitations must be considered when extrapolating to humans. This applies to the hyperbaric chamber diving protocol and also to the amount of CO_2_, which may seem extreme. However, this allows reliable clinical signs to be produced in the animals.

In this study, it was unfortunate that before-and-after comparisons could not be made for blood analysis and fecal bacterial identification. Exposure to CO_2_ significantly reduced the number of asymptomatic animals that could be sampled. Accidentology reduced blood perfusion (access to punction) and also fecal excretion.

In fact, we have favored the symptomatology to the detriment of the technique of ultrasound bubble detection in this animal. Clinical evaluation remains essential, as it is well known that the presence of bubbles does not necessarily correlate with the reporting of DCS symptoms. We assume that we did not quantify circulating bubbles in this study, whereas their diameter is influenced by gases dissolved in the tissue, such as CO_2_ [90]. Because we cannot exclude the involvement of CO_2_ in the growth of the bubbles, especially in symptomatic mice previously exposed to CO_2_, this point deserves clarification.

We mention the influence of the genetic background of the mice on the selection of the microbiota. To the extent that we have used a single animal strain, it would be interesting to combine several strains to verify their impact on microbial communities and accidentology, and then try to identify a profile, as could be done in rats [25,31,80].

Regarding the microbiota, the low identification rate could be a limitation of this study. Approximately 97% (min 4%–max 99%) of the ASVs were not classified/identified, which requires a cautious interpretation of these results. This remaining 97% could represent, for example, microbes, but also their degradation products or even food waste. Let us recall that this is a feces analysis.

Certain bacterial strains have emerged from this work. This study is not specifically dedicated to their identification or their mode of action, so caution should be exercised in their interpretation. This work could be directed by the use of antibiotics or, on the contrary, by carrying out microbiota transplants, for example.

## 5. Conclusions

This work is the first to clearly demonstrate the influence of exposure to a high concentration of CO_2_ on DCS risk. The exposure of mice to a maximum CO_2_ concentration of 70 hPa for one hour prior to the hyperbaric protocol dramatically increased the incidence of DCS. This was confirmed by temperature measurements at the end of the procedure. More unexpectedly, it seems that the lower temperature of the animals even before exposure to the accident-prone protocol leads to an unfavorable prognosis. This study also suggests that the microbiota may partially influence thermogenesis and thus accidentology. For example, the low representation of *Lachnospiraceae_GCA-9000066575* could be perceived as protective in the air condition. However, depending on prior exposure, certain high abundances of bacteria identified in this work, such as *Lachnospiraceae_Blautia*, could be perceived as favoring DCS or, on the contrary, as protective if the risky dive follows exposure to CO_2_. Studies should further explore this issue in order to improve prevention in diving.

## Figures and Tables

**Figure 1 ijerph-21-01141-f001:**
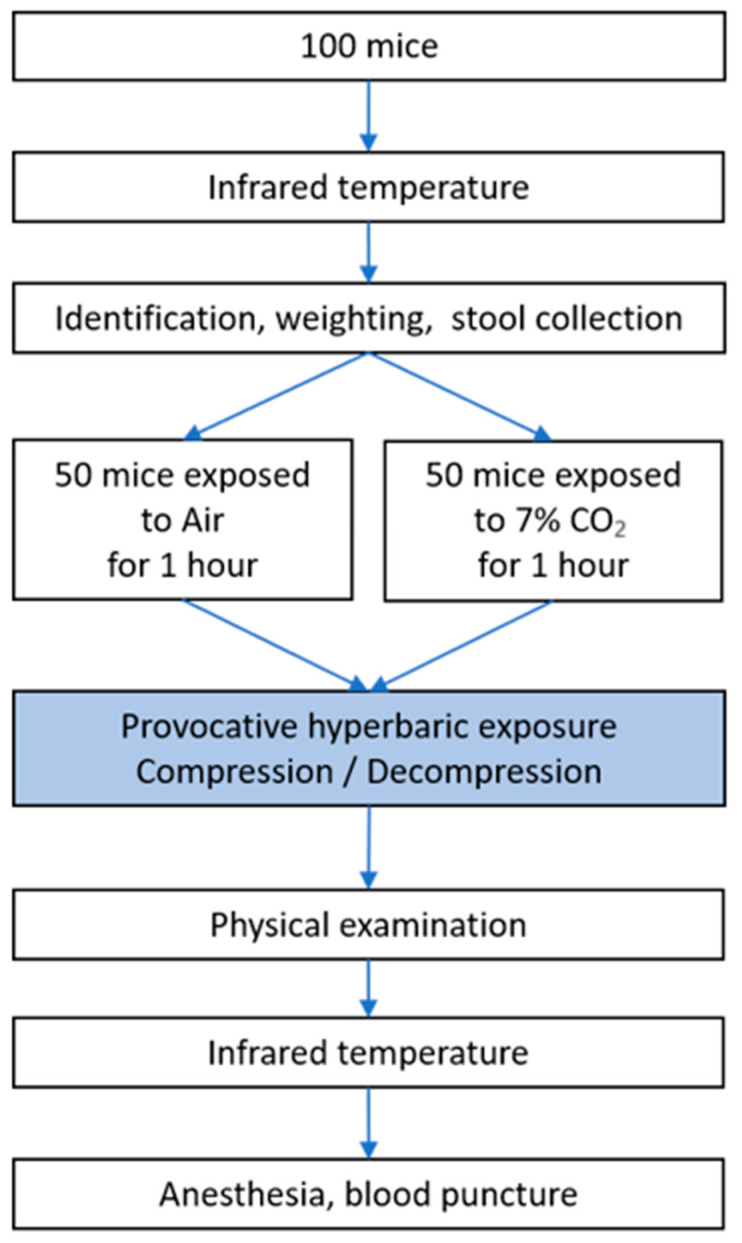
Flow chart of experimental design—measurement of temperature, identification, weighing, and feces collection for microbiota analysis in mice.

**Figure 2 ijerph-21-01141-f002:**
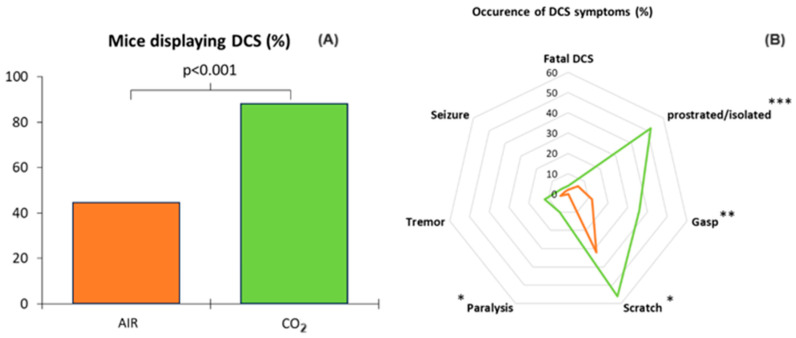
Decompression sickness occurrences 30 min after pathogenic decompression as a function of previous gas exposure (AIR or CO_2_). (**A**) **Histogram**: Proportion of mice with DCS symptoms. The DCS status was attributed when the mice presented neurological signs in the form of paresis or paralysis of at least one limb, convulsions, and/or reduced performance in the motor test for forelimbs, and also respiratory troubles, and even death. (**B**) **Radar Chart**: Percentage of mice displaying a type of symptom in a population, considering that a mouse can present the whole panel of symptoms. * denotes *p* < 0.050, ** denotes *p* < 0.010, *** denotes *p* < 0.001.

**Figure 3 ijerph-21-01141-f003:**
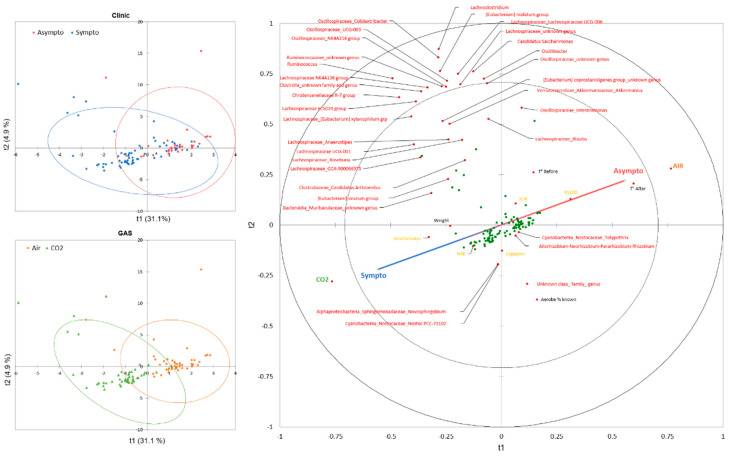
Partial Least Squares Discriminant Analysis (PLS-DA) plot conducted on clinical status (symptomatic or asymptomatic, blue and red gradient line) as a dependent variable and as explanatory variables, gas exposure before the hyperbaric protocol (CO_2_, green or air, brown), micro-organisms genus detected in stools (red), mice superficial temperature before and after the protocol (black), and plasma analysis performed at the end of the protocol (orange). (**Right**) Correlation map based on observations (green dots) according to their qualitative or quantitative explanatory characteristics. (**Left**) Confusion matrix summarizing the observations according to the clinical status of the mice (symptomatic in blue or asymptomatic in red), according to the gas breathed before the dive (air in brown or CO_2_ in green), with their confidence ellipse (Fisher, confidence interval 95%).

**Figure 4 ijerph-21-01141-f004:**
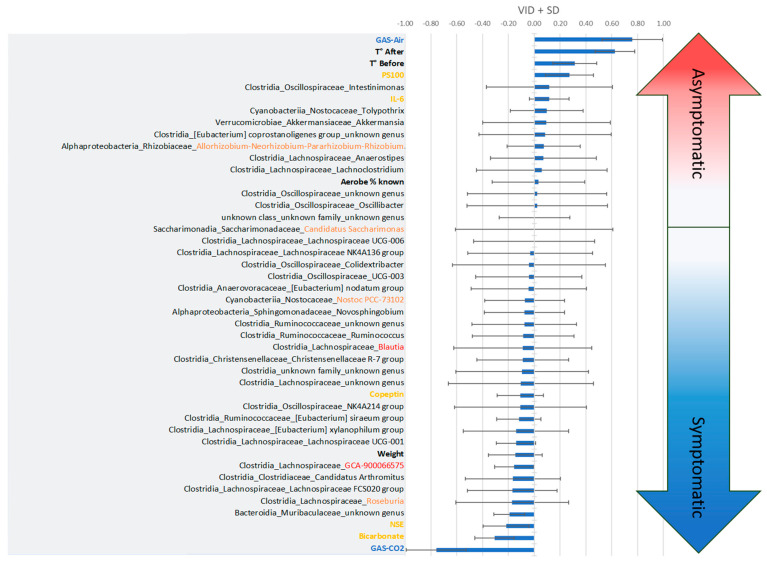
Classification of the variables of importance in projection according to the discrimination (VIDs), which specify the interest of the parameters according to the clinical state. According to PLS-DA, clinical trend arrows appear in gradient color. The more positive the values (top), the closer they are to the asymptomatic clinical status. The more negative the values (bottom), the more the mice expressed DCS symptoms. VIDs ± standard deviation.

**Figure 5 ijerph-21-01141-f005:**
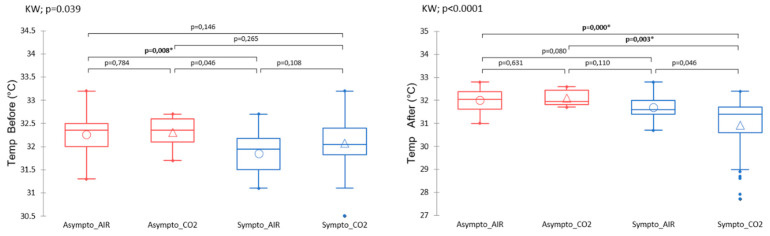
Surface temperature measured by infrared (from thermograms) at the very first manipulation, or after the hyperbaric protocol. The clinical group is represented by colors, and the gas group is represented by shape. Please refer to Figure 1 for the tracking timeline. Kruskal–Wallis (KW) analysis with post hoc Dunn test (Bonferroni correction: 0.0083). * denotes *p* < 0.050.

**Figure 6 ijerph-21-01141-f006:**
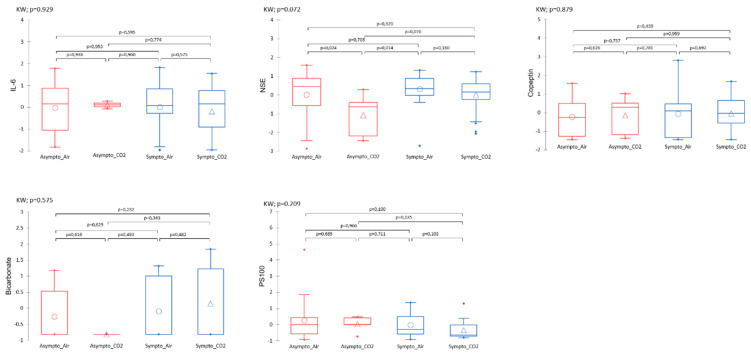
Plasmatic analyses carried out at the very end of the protocol, i.e., after the provocative hyperbaric protocol. The clinical group is represented by colors, and the gas group is represented by shape. Please refer to Figure 1 for the tracking timeline. Normalized data. Kruskal–Wallis (KW) analysis with post hoc Dunn test (Bonferroni correction: 0.0083).

**Figure 7 ijerph-21-01141-f007:**
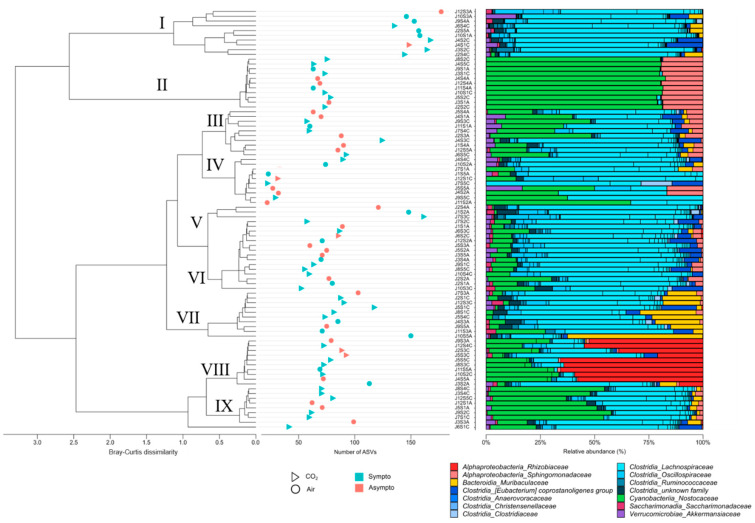
The dendrogram clustering illustrates the Bray–Curtis dissimilarity of the taxonomic profile, which was performed using the ASV abundance table. The scale bar of the dendrogram represents the degree of dissimilarity (%) between the prokaryotic communities. The number of ASVs was calculated from the 16S rDNA ASV table for each sample. The clinical group is represented by colors, and the gas group is represented by shape. The bar chart illustrates the cluster from the Bray–Curtis dendrogram. Each bar represents the relative abundance of the family in the sample under study.

**Figure 8 ijerph-21-01141-f008:**
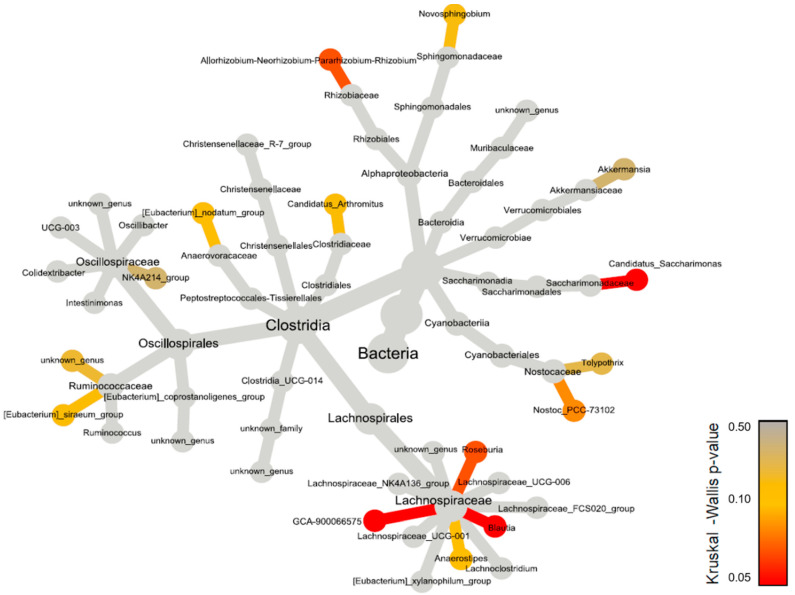
Heat tree constructed using 16S rDNA data at the genus level. Each connected node represents a different taxonomic rank. The nodes are colored based on the Kruskal–Wallis differences between the 4 conditions (Asympto_AIR, Asympto_CO_2_, Sympto_AIR and Sympto_CO_2_).

## Data Availability

The results of this study have been deposited in the NCBI Sequence Read Archive under the accession number BioProject PRJNA1070862.

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
