# Peer review of "CO2 Breathing Prior to Simulated Diving Increases Decompression Sickness Risk in a Mouse Model: The Microbiota Trail Is Not Forgotten"

_ijerph, 2024, doi:10.3390/ijerph21091141_

Round 1

Reviewer 1 Report

Comments and Suggestions for Authors

This study provides valuable insights into the relationship between CO2 exposure, body temperature, and DCS risk in mice. The dramatic increase in DCS incidence following CO2 exposure is a significant finding that could have important implications for diving safety protocols.

The text does not provide details on the provocative dive protocol, which is crucial for understanding the experimental design.

The mechanism by which CO2 exposure leads to increased DCS incidence is not fully explained.

The study is conducted on mice, which may limit direct applicability to human divers.

References are missing in the statistical analysis section.

Author Response

This study provides valuable insights into the relationship between CO2 exposure, body temperature, and DCS risk in mice. The dramatic increase in DCS incidence following CO2 exposure is a significant finding that could have important implications for diving safety protocols.

The authors would like to thank you for this proofreading work. We have tried to meet your expectations as well as those of your colleagues. A document with the tracking of corrections should be available. We remain at your disposal. Kind regards

The text does not provide details on the provocative dive protocol, which is crucial for understanding the experimental design.
We hope you'll find your answers in chapter 2.5 “Hyperbaric protocol”. If not, please don't hesitate to specify your requirements. We'll do our best to meet them.

The mechanism by which CO2 exposure leads to increased DCS incidence is not fully explained.
This study was not specifically dedicated to elucidating the mechanisms of CO2. We understand your expectations, but would have been too subjective to state them without further proof. Line 45, we added our recent review published on the subject: Effects of COâ‚‚ on the occurrence of decompression sickness: review of the literature. Daubresse L, Vallée N, Druelle A, Castagna O, Guieu R, Blatteau JE. 2024.

The study is conducted on mice, which may limit direct applicability to human divers.

This is an usual limit of studies conducted on animals, as explained in chapter 4.6 Moderation: “The mouse is probably not the best model for DCS, and as with any model, its limitations must be considered when extrapolating to humans. This applies to the hyperbaric chamber diving protocol and also to the amount of CO2, which may seem extreme. However, this allows reliable clinical signs to be produced in animals.”

References are missing in the statistical analysis section.

References, particularly on the PLS-DA, have been added to the text.

Reviewer 2 Report

Comments and Suggestions for Authors

Thank you for the opportunity to review this manuscript which is of interest to the research community involved in DCS pathophysiology research.
The authors investigate the effect of CO2 prebreathing and it’s impact on DCS outcome on a mice model. This study is technically well conducted, and has good potential to contribute to the field of baromedicine. I believe that the authors have interesting data to report and that they can improve the manuscript through revision, in particular by correcting the English and making slight improvements.

General comments:

-          a last thorough proof reading might help to correct minor typos (lines 38 and 40 CO2is, ref line 272, line 363 and 364 axis 1 or t1, line 491 a space is missing, line 578  A1>, etc.)

-          some abbreviations are not explained on their first occurrence and there are many of them, so I recommend to limit their use to the minimum to facilitate the read.

-          Some figures are hard to read when we print the manuscript and might be easier to read if enlarged

Title.

Line 16 remove one of the correspondence.

Abstract.

Lines 20-21: neurological disorder might be the leading cause of dive accidents at the hyperbaric chamber, but ear issues remain more common. Maybe you can rephrase to neuro DCS is the leading cause of major diving accidents treated in hyperbaric chambers?

Introduction.

Line 57 -58: can you develop the correlation between circulating bubbles and DCS? And what about tissue bubbles?

Line 67: the most common symptoms… treated at the hyperbaric chamber?

Line 70: the leading cause of major diving accidents?

Lines 78 to 81: can you develop why?

Line 94: your team? Anyone else?

Mat & Meth.

Line 133: Details follow. Where?

Can you comment the comparison with real life diving? You are explaining the context at the end, but maybe you can move it to the introduction? Why CO2 exposure before the dive and not during and/or after the dive?

General comment: how long were the animals stabulated before the experiment? Did you conduct the experiments at any specific time of the day to limit any chronobiological impact?

Line 182: is it really 1 atm/s or 1 atm/min?

Results.

Line 305 and 306? Why was it expected? I think moving the context in the introduction will help to better understand your study design.

Line 366 and 367: you previously selected the animal base on their weight in a previous study or for this one? What’s the repartition between the groups?

Discussion.

Line 528: if this is the first study to demonstrate the effect of CO2 before the dive, why did you expect these results?

Paragraph 4.5: I think you will gain in clarity if you move it in the introduction.

Lines 719 to 721: can you reformulate the approximately 97% and remaining 97% as I am not sure to understand what you mean?

Data availability statement: Please fill this statement.

Comments on the Quality of English Language

The manuscript will benefit a thorough proofreading to correct some typos and expressions that can be improved.

Author Response

Thank you for the opportunity to review this manuscript which is of interest to the research community involved in DCS pathophysiology research.
The authors investigate the effect of CO2 prebreathing and it’s impact on DCS outcome on a mice model. This study is technically well conducted, and has good potential to contribute to the field of baromedicine. I believe that the authors have interesting data to report and that they can improve the manuscript through revision, in particular by correcting the English and making slight improvements.

 The authors would like to thank you for this valuable proofreading work. We have tried to meet your expectations as well as those of your colleagues. A document with the tracking of corrections should be available. We remain at your disposal. Kind regards

General comments:

-          a last thorough proof reading might help to correct minor typos (lines 38 and 40 CO2is, ref line 272, line 363 and 364 axis 1 or t1, line 491 a space is missing, line 578  A1>, etc.)

Some typos are from our version and we apologize for them. Others are from the layout of the journal. We have corrected as many as possible.

-          some abbreviations are not explained on their first occurrence and there are many of them, so I recommend to limit their use to the minimum to facilitate the read.

OK

-          Some figures are hard to read when we print the manuscript and might be easier to read if enlarged

This is a problem beyond our control. It must be an editing problem because our figures are in high definition.

Title.

Line 16 remove one of the correspondence.

Done

Abstract.

Lines 20-21: neurological disorder might be the leading cause of dive accidents at the hyperbaric chamber, but ear issues remain more common. Maybe you can rephrase to neuro DCS is the leading cause of major diving accidents treated in hyperbaric chambers?

 That’s ok

Introduction.

Line 57 -58: can you develop the correlation between circulating bubbles and DCS? And what about tissue bubbles?

We have reworded the first sentence and added a second one mentioning the existence of stationary bubbles, although the latter are less often cited in the literature as being the cause of neurological DCS.

« According to the classical scheme, DCS results from circulating bubbles in the bloodstream » ==>

“According to the classical scheme, DCS results from the formation of bubbles in the body….. inefficient. Bubbles that “remain trapped” in tissues exert their effects primarily through compression of their surroundings. In the bloodstream, the presence of bubbles (i.e. vascular gas embolism) causes cell destruction ……

Line 67: the most common symptoms… treated at the hyperbaric chamber? And Line 70: the leading cause of major diving accidents?

The wording was haphazard and even confusing. We removed the last sentence and removed the “most common” part in favor of “most serious”…. treated at the hyperbaric chamber.

Lines 78 to 81: can you develop why?

We added a new reference with the aim to introduce the counterdiffusion notion, briefly.

Line 94: your team? Anyone else?

Actually not, references are sorely lacking (Lautridou et al…).

Mat & Meth.

Line 133: Details follow. Where?

We suggest “See details in the following paragraphes.”

Can you comment the comparison with real life diving? You are explaining the context at the end, but maybe you can move it to the introduction? Why CO2 exposure before the dive and not during and/or after the dive? Paragraph 4.5: I think you will gain in clarity if you move it in the introduction.

Although we understand the interest of your remark, and that a reference is made to the context in the second sentence of the introduction, it seems delicate to us to highlight further the (speculative) consequences of such an exposure without having first presented the results of this study carried out on animals.

General comment: how long were the animals stabulated before the experiment?

2 weeks

Did you conduct the experiments at any specific time of the day to limit any chronobiological impact?

Yes. it started at 9am

Line 182: is it really 1 atm/s or 1 atm/min?

 Typo. It was 0.1 atm/s.

Results.

Line 305 and 306? Why was it expected? I think moving the context in the introduction will help to better understand your study design. And Line 366 and 367: you previously selected the animal base on their weight in a previous study or for this one? What’s the repartition between the groups?

Weight affects the DCS rate. The heavier the animal, the higher the risk of DCS. We hypothesized an increased risk under the effect of CO2. However, we did not want to have 0% mortality to validate the DCS, nor 100% mortality in order to have access to blood tests among other things. We choose to order lighter animals. This is what we expected.

As previously precised in the text line 327, no significant difference was found between groups. We added the mean weight in the text (19.2 ± 1.7g).

Discussion.

Line 528: if this is the first study to demonstrate the effect of CO2 before the dive, why did you expect these results?

We added « To our knowledge, this study is the first to demonstrate… in mice ». We recently wrote a short review on the subject: Daubresse et al 2024.

Lines 719 to 721: can you reformulate the approximately 97% and remaining 97% as I am not sure to understand what you mean?

This is in the study limits section. It refers to the sequencing results part: Line 464: “….16s rDNA analysis was performed on mouse feces collected at the very beginning of the protocol. Approximately 97% (min 4%_max 99%) of ASVs were not classified/identified, so these results must be interpreted with caution (see Limitations section). Finally, 520 ASVs from the stools of 88 mice allowed the identification of about thirty genera, distributed in about 13 families and 6 classes, with the presence of unknowns….. ». that could be surprising.

Actually, There is a typo : « these » should be prefered to « The ».

« …these results. These remaining 97% could represent, for example, microbes, but also their degradation products or even food waste … » We also remind (add in the text) that this is a fecal analysis.

In fact, the next step of NGS will be to easily neutralize the host or environmental DNA that creates too much background noise, including by applying mathematical filters. Here, by applying these rather severe filters, we are left with “only” 3% of identified ASVs, which still represent 520 ASVs.

Data availability statement: Please fill this statement.

This should be an automatic section…!? Actually, this study have been deposited in the NCBI Sequence Read Archive under the accession number BioProject PRJNA1070862, as precised line 283. We change it.

Comments on the Quality of English Language

The manuscript will benefit a thorough proofreading to correct some typos and expressions that can be improved.

Thank you very much

Reviewer 3 Report

Comments and Suggestions for Authors

Interesting and necessary research. Its objective is: The purpose of this study was to evaluate the effect of CO2 breathing prior to a provocative dive on the occurrence of DCS in mice.

The authors are requested to address the following items:

1)      Introduce basic statistical data in the summary section.

2)      Introduce the research objective in the last paragraph of the introduction section. Likewise, start the description of the research objective in the first paragraph of the discussion section.

3)      In the methods section (2.11), normality is declared (Lines: 286-87). However, non-parametric statistics are used (Kruskal-Wallis tests: Line: 298). Use the correct statistic for n independent samples according to the existing normality.

Author Response

The authors would like to thank you for this valuable proofreading work. We have tried to meet your expectations as well as those of your colleagues. A document with the tracking of corrections should be available. We remain at your disposal. Kind regards

The authors are requested to address the following items:

1)      Introduce basic statistical data in the summary section.

This is done.

2)      Introduce the research objective in the last paragraph of the introduction section. Likewise, start the description of the research objective in the first paragraph of the discussion section.

You are right. It was necessary. So we added these sentences in the introduction and in the discussion.

3)      In the methods section (2.11), normality is declared (Lines: 286-87). However, non-parametric statistics are used (Kruskal-Wallis tests: Line: 298). Use the correct statistic for n independent samples according to the existing normality

We have rephrased this part.

Thank you

Reviewer 4 Report

Comments and Suggestions for Authors

Comments

I have reviewed the article "CO2-Breathing Prior to Simulated Diving Increases Decompression Sickness Risk in a Mouse Model: The Microbiota Trail Is Not Forgotten" and found some flaws that could be rectified during revision.

The word “correspondence” appears two time; please remove one.

Line 26. As the atmospheric pressure changes with altitudes, it is necessary to mention the place (the city where the experiment was performed, if possible); 7% at atmospheric pressure in line 120 can be copied and pasted in the abstract.

Line 28. (controls) or (control)?

Line 27. In the AIR group…or air group?

Line 36. Protective or pathogenic, I suggest modifying this with “beneficial or pathogenic.” However, I couldn't find any beneficial and pathogenic bacteria in the whole manuscript except for bacteria genera or families.

Keywords: The authors have added eight Keyword words, which must be limited to 4-5 words.

Introduction

The first and third lines “CO2is” please add space as CO2 is…, In addition, please revise the whole text to correct such typing errors.

Line 69. Please start with DCS instead of Decompression sickness, as DCS appears before.

Line 126. Which temperature was measured, body temperature or else?

Line 127-127. Mice were then immediately identified… please clarify the term “identified”.

Line 235-243. Please split this paragraph into 2-4 sentences, which is too long to convey the proper meaning.

Line 314. AIR group or air group? The bar graph in Figure 2 shows air, not AIR. Please check the whole text for such corrections if necessary.

Line 310, 632, etc., if any. Please add DCS (already abbreviated) instead of decompression sickness.

Line 470-471. Only about fifteen species appeared correctly identified, prompting analyses at the genus level. Which species? Can the author name these species? I couldn't find them in the manuscript text.

The rest of the article is ok.

Author Response

The authors would like to thank you for this valuable proofreading work. We have tried to meet your expectations as well as those of your colleagues. A document with the tracking of corrections should be available. We remain at your disposal.

Kind regards

The word “correspondence” appears two time; please remove one.

Done

Line 26. As the atmospheric pressure changes with altitudes, it is necessary to mention the place (the city where the experiment was performed, if possible);

This sentence was added at the very beginning of the Materials chapter. “This study was conducted in Marseille (France)”

7% at atmospheric pressure in line 120 can be copied and pasted in the abstract.

Done

Line 28. (controls) or (control)?

The control group

Line 27. In the AIR group…or air group?

AIR for group and air for the gas.

Line 36. Protective or pathogenic, I suggest modifying this with “beneficial or pathogenic.” However, I couldn't find any beneficial and pathogenic bacteria in the whole manuscript except for bacteria genera or families.

You are right. This is more suitable. We have added the word "genera" in the abstract, to remain in line with the results presented.

Keywords: The authors have added eight Keyword words, which must be limited to 4-5 words.

OK

Introduction

The first and third lines “CO2is” please add space as CO2 is…, In addition, please revise the whole text to correct such typing errors.

OK

Line 69. Please start with DCS instead of Decompression sickness, as DCS appears before.

OK

Line 126. Which temperature was measured, body temperature or else?

OK. The surface temperature. Details are provided in 2.3

Line 127-127. Mice were then immediately identified… please clarify the term “identified”.

It is added. The subjects were identified by “a coloured barcode affixed to the base of the tail” via 3 fine lines and 3 colors.

Line 235-243. Please split this paragraph into 2-4 sentences, which is too long to convey the proper meaning.

Ok this is done: « Bicarbonate (HCO3-) was determinated with a volume of 5.7 µL per assay on Atellica Solution (Siemens®) platform equipped with the CH930 module (detection limit of 10 µmol/L). The assay uses a colorimetric enzymatic method based on two chemical reactions, catalysed by phosphoenolpyruvate carboxylase (PEPC) and malate dehydrogenase (MDH). The reduction in absorbance is proportional to the concentration of the analyte. Similarly, IL-6 (50 µL per assay / detection limit 2.7 pg/mL) is determined on the same platform with the IM module, which employs chemiluminescence using a mouse monoclonal antibody labelled with an acridinium ester molecule and monoclonal antibody coated magnetic microparticles, both directed against IL-6.”

Line 314. AIR group or air group? The bar graph in Figure 2 shows air, not AIR. Please check the whole text for such corrections if necessary.

OK, fig 2 revised

Line 310, 632, etc., if any. Please add DCS (already abbreviated) instead of decompression sickness.

Line 470-471. Only about fifteen species appeared correctly identified, prompting analyses at the genus level. Which species? Can the author name these species? I couldn't find them in the manuscript text.

Here is the list of « identified » species: Akkermansia muciniphila, Candidatus Arthromitus sp. SFB-mouse-Japan, Clostridium sp., Dorea sp., Lachnospiraceae bacterium 10-1, Lachnospiraceae bacterium 28-4, Nostoc punctiforme PCC 73102, Nostoc sp., Novosphingobium resinovorum, Rhizobium sp., bacterium YRD2003

There are also even less well identified species, affiliated with gut metagenome, metagenome, mouse gut metagenome, unidentified rumen bacterium JW32, and clearly unidentified species.

The rest of the article is ok

Thank you
